# Generation and Characterization of Trastuzumab/Pertuzumab-Resistant HER2-Positive Breast Cancer Cell Lines

**DOI:** 10.3390/ijms25010207

**Published:** 2023-12-22

**Authors:** Marta Sanz-Álvarez, Melani Luque, Miriam Morales-Gallego, Ion Cristóbal, Natalia Ramírez-Merino, Yamileth Rangel, Yann Izarzugaza, Pilar Eroles, Joan Albanell, Juan Madoz-Gúrpide, Federico Rojo

**Affiliations:** 1Department of Pathology, Fundación Jiménez Díaz University Hospital Health Research Institute (IIS—FJD, UAM)—CIBERONC, 28040 Madrid, Spain; marta.sanza@quironsalud.es (M.S.-Á.); melani.luque@quironsalud.es (M.L.); miriam.moralesg@quironsalud.es (M.M.-G.); 2Translational Oncology Division, OncoHealth Institute, Fundación Jiménez Díaz University Hospital Health Research Institute (IIS—FJD, UAM)—CIBERONC, 28040 Madrid, Spain; ion.cristobal@quironsalud.es; 3Department of Medical Oncology, Infanta Elena University Hospital, 28342 Madrid, Spain; natalia.ramirezm@quironsalud.es; 4Department of Pathology, Infanta Elena University Hospital, 28342 Madrid, Spain; yamileth.rangel@quironsalud.es; 5Department of Medical Oncology, Fundación Jiménez Díaz University Hospital, 28040 Madrid, Spain; yizarzugaza@quironsalud.es; 6Institute of Health Research INCLIVA—CIBERONC, 46010 Valencia, Spain; pilar.eroles@uv.es; 7Department of Physiology, University of Valencia, 46010 Valencia, Spain; 8Cancer Research Program, IMIM (Hospital del Mar Research Institute), 08003 Barcelona, Spain; 96087@parcdesalutmar.cat; 9Department of Medical Oncology, Hospital del Mar—CIBERONC, 08003 Barcelona, Spain; 10Department of Experimental and Health Sciences, Faculty of Medicine, Universitat Pompeu Fabra, 08002 Barcelona, Spain

**Keywords:** breast cancer, HER2 positive, targeted therapy, trastuzumab, pertuzumab, resistance, label-free proteomics

## Abstract

The combination of trastuzumab and pertuzumab as first-line therapy in patients with HER2-positive breast cancer has shown significant clinical benefits compared to trastuzumab alone. However, despite initial therapeutic success, most patients eventually progress, and tumors develop acquired resistance and invariably relapse. Therefore, there is an urgent need to improve our understanding of the mechanisms governing resistance in order to develop targeted therapeutic strategies with improved efficacy. We generated four novel HER2-positive cell lines via prolonged exposure to trastuzumab and pertuzumab and determined their resistance rates. Long-term resistance was confirmed by a significant increase in the colony-forming capacity of the derived cells. We authenticated the molecular identity of the new lines via both immunohistochemistry for the clinical phenotype and molecular profiling of point mutations. HER2 overexpression was confirmed in all resistant cell lines, and acquisition of resistance to trastuzumab and pertuzumab did not translate into differences in ER, PR, and HER2 receptor expression. In contrast, changes in the expression and activity of other HER family members, particularly HER4, were observed. In the same vein, analyses of the receptor and effector kinase status of different cellular pathways revealed that the MAPK pathway may be involved in the acquisition of resistance to trastuzumab and pertuzumab. Finally, proteomic analysis confirmed a significant change in the abundance patterns of more than 600 proteins with implications in key biological processes, such as ribosome formation, mitochondrial activity, and metabolism, which could be relevant mechanisms in the generation of resistance in HER2-positive breast cancer. We concluded that these resistant BCCLs may be a valuable tool to better understand the mechanisms of acquisition of resistance to trastuzumab and pertuzumab-based anti-HER2 therapy.

## 1. Introduction

Breast cancer is the most frequently diagnosed cancer across the globe, and it accounted for 2.3 million new cases and 685,000 deaths in 2020 [1]. Furthermore, it is the most prevalent form of cancer worldwide. Breast cancer is a heterogeneous disease with varying prognoses that rely on different histological and molecular characteristics [2]. Breast carcinomas are classified into four major molecular subtypes, largely defined by the expression of hormonal receptors (estrogen and progesterone) and HER2 receptor: luminal A, luminal B, HER2 positive, and triple negative or basal like. The HER2-positive subtype, which accounts for approximately 25% of breast cancers, is characterized by either HER2 gene amplification or protein overexpression [3]. It has been associated with aggressive behaviors and reduced disease-free and overall survival [4,5]. The HER2 receptor is one of the four transmembrane tyrosine kinase receptors of the ErbB family, which also includes the EGFR, HER3, and HER4 [6]. These receptors interact among themselves, whether or not they bind to a ligand, leading to the activation of major signaling pathways like the PI3K/AKT, MAPK, and JAK/STAT pathways, which regulate various cellular processes like cell growth, differentiation, motility, and invasion [7,8].

The development of targeted therapies against HER2 has significantly improved the survival and quality of life for HER2-positive breast cancer patients. Specifically, trastuzumab and pertuzumab, humanized monoclonal antibodies that bind to different domains of the HER2 receptor, have demonstrated efficacy in clinical trials and are currently approved for use in combination with chemotherapy as the standard of care for HER2-positive breast cancer. Both antibodies have complementary mechanisms of action: trastuzumab inhibits HER2 dimerization, whereas pertuzumab impedes HER2 heterodimerization with other HER family receptors, especially HER3 [9,10]. These drugs have been shown to inhibit HER2 signaling, induce antibody-dependent cellular cytotoxicity, and improve overall survival and progression-free survival in both early and metastatic HER2-positive breast cancer patients [11,12]. Trastuzumab was approved by the FDA in 1998 for the treatment of early-stage and metastatic HER2-positive breast cancer patients [13,14]. Subsequently, in 2013, pertuzumab was granted approval for use in combination with trastuzumab and chemotherapy as a therapeutic option for patients with HER2-positive disease [11,15].

Despite the considerable improvement in the management of HER2-positive breast cancer patients using targeted therapies, the majority of patients who initially respond eventually develop resistance to these therapies within a year. Over the past few years, several mechanisms of resistance to trastuzumab therapies have been proposed, including the expression of truncated forms of the HER2 receptor, such as p95HER2 [16] or HER2Δ16 [17]; increased activity of ligands and other receptors in the HER signaling network, especially HER3 [18]; activation of alternative signaling pathways, such as that of the insulin-like growth factor I receptor [19] (IGF-IR); aberrant activation of the PI3K/AKT or MAPK signaling pathways [20,21]; deregulation of metabolism [22,23]; or immune-mediated mechanisms, such as the overexpression of programmed death-ligand 1 [24]. While less studied, resistance to pertuzumab as a monotherapy has also been reported, with studies suggesting mechanisms linked to activating mutations [25], the formation of EGFR–HER3 heterodimers [26], or even miRNA regulation [27]. Dual resistance to trastuzumab and pertuzumab is an area where few studies have described the mechanisms involved. Recent studies have suggested that HER3 may be implicated, as demonstrated by the activation of HER2–HER3 in in vivo models and the possible use of HER2 kinase inhibitors [28]. Additionally, when mutated, PIK3CA may play a role in the development of resistance, as indicated in several mouse models that have confirmed metastasis progression and the emergence of resistance to trastuzumab and pertuzumab [29,30]. Therefore, it is essential to identify new mechanisms of resistance to both antibodies to anticipate patients’ relapse and improve response rates. The generation of resistant cell lines via continuous drug exposure would enable the elucidation of the mechanisms involved in resistance emergence, the identification of novel biomarkers to improve patients’ management, and, ultimately, the discovery of new treatment approaches for HER2-positive breast cancer patients.

Since proteins play a crucial role in cellular processes, including metabolism, DNA replication and repair, cell signaling, and the immune response [31], they are the main targets for drug development [32]. Moreover, proteins can be used as biomarkers for drug efficacy, safety, and toxicity. Overall, the study of the role of proteins in drug studies is critical for developing safe and effective treatments for a wide range of cancers. For this reason, proteomics has emerged as an important tool for protein analyses in cancer research, and specifically in breast cancer [33,34]. It has provided a deeper understanding of the changes in protein expression and function in cancer cells, which has helped to identify potential drug targets and improve therapies. Furthermore, proteomic studies have helped to elucidate the mechanisms of drug resistance in cancer cells, providing opportunities for the development of novel therapeutic approaches. Overall, proteomic studies offer valuable insights into the complex molecular mechanisms of cancer and provide critical information for the development of more effective treatments.

This study aimed to develop and characterize four cellular models resistant to dual anti-HER2 blockades with trastuzumab and pertuzumab. The main objective was to evaluate the efficacy of these models as tools to understand the underlying mechanisms of resistance to these anti-HER2 therapies. The characterization of these cell lines included mutational analyses and the assessment of their phenotype to authenticate their validity as models. Additionally, we examined the expression and activation status of HER family receptors, as well as the PI3K/AKT and MAPK signaling pathways, which have been previously reported to be modulated by the therapies employed. Finally, we conducted label-free mass spectrometry proteomic analysis to identify and quantify potential biomarkers that may be involved in the acquisition of resistance to trastuzumab and pertuzumab.

## 2. Results

### 2.1. Generation of HER2-Positive BCCLs with Acquired Resistance to Trastuzumab and Pertuzumab

Based on our previous findings and reports in the literature [35,36,37], we generated in vitro cellular models of acquired resistance to both trastuzumab and pertuzumab in four well-established HER2-positive BCCLs: AU-565, BT-474, EFM-192A, and SK-BR-3. Acquired resistance was generated via continued exposure to a combination treatment of 15 µg/mL trastuzumab and 20 µg/mL pertuzumab for periods ranging from 8 months (AU-565, EFM-192A, and SK-BR-3) to 15 months (BT-474). These concentrations were chosen based on the existing scientific literature and the previous experience of our group. The concentrations found in both samples were deemed to be comparable to those observed in patients treated with these drugs during clinical trials [37,38,39,40] and in preclinical models [36,41,42,43]. This replicates the concentration present during the development of resistance in patients. As considered in previous reports, parental cell lines were deemed sensitive and were cultured in parallel without treatment as a control. Resistance was evaluated monthly through proliferation assays in the presence and absence of treatment. While all the parental cell lines were sensitive to both therapies, the generated resistant models (named as “cell line name.rTP” hereafter) exhibited proliferation rates above 80% in the presence of trastuzumab and pertuzumab treatment following their exposure times ranging from 8 to 15 months (Figure 1A–D). For each resistant cell line, we generated three different pools and selected those exhibiting the highest percentage of resistance. Subsequently, single-cell clones were isolated from the selected pools of AU-565.rTP-, EFM-192A.rTP-, and SK-BR-3.rTP-resistant cell lines via limited dilution cloning. The following experiments were performed using the corresponding clone that had the highest resistance rate. To ensure consistency, we used the same nomenclature for all the lines, whether they were clones or pools. No morphological changes were observed in any of the HER2-positive cell lines that were used when cell lines with acquired resistance to trastuzumab and pertuzumab were generated.

The response to trastuzumab and pertuzumab was quantified by calculating the change in the growth rate of treated versus non-treated cells, according to the algorithm described by O’Brien et al. [44]. Consequently, cell lines with a growth rate-fold increase in FC ≥ 1.2 were considered resistant to both therapies (Table 1). The acquisition of resistance was monitored monthly via cell proliferation assays, and upon confirmation of resistance, the treatment was halted for 30 days.

In order to evaluate the resistance to trastuzumab and pertuzumab over periods exceeding 7 days (long-term resistance), a clonogenic assay was employed. As the periods of exposure to these drugs extended over weeks under conditions that exceeded the simulation of clinical treatment, we lowered the drug concentrations to minimize their potential cytotoxic effects. Following 21 days of treatment with 1 µg/mL trastuzumab and 1.3 µg/mL pertuzumab, the relative number of colonies was compared between the sensitive and resistant cell lines (Figure 2A–D). As such, a significant increase in the number of colonies in the presence of the treatment was observed between the sensitive and resistant cells in all the evaluated cell models (Figure 2E–H). Notably, the BT-474 and BT-474.rTP cell lines exhibited major differences in their number of colonies after combined treatment for 21 days (7% vs. 97%, respectively). Additionally, differences in the size and number of colonies were identified between the sensitive cell lines AU-565 and EFM-192A and their resistant counterparts.

### 2.2. Acquiring Resistance Did Not Result in Any Alterations to the Molecular Profile of the BCCLs

All the BCCLs, sensitive and resistant to dual anti-HER2 therapy, underwent molecular characterization using surrogate markers for clinical subtype classification. Positive nuclear staining for estrogen receptor (ER) and progesterone receptor (PR) was observed in BT-474, BT-474.rTP, EFM-192A, and EFM-192A.rTP (Figure 3B,C), while no nuclear staining was detected for either of these receptors in the SK-BR-3, SK-BR-3.rTP, AU-565, or AU-565.rTP cell lines (Figure 3A,D). Overexpression of HER2 (3+) was detected via immunohistochemistry (IHC) in all the cell lines that were examined (Figure 3A–D). Based on these criteria, our cell models were classified as “luminal” if ER, PR, and HER2 expression were detected (BT-474 and EFM-192A) or “HER2-positive” if only HER2 expression was identified (AU-565 and SK-BR-3). Acquisition of resistance to trastuzumab and pertuzumab did not result in differences in ER, PR, and HER2 receptor expression.

Cell line authentication was performed to prevent misidentification and ensure accurate categorization of the new cellular models generated. A qPCR assay was developed to discriminate between the different point mutations evaluated in the BCCLs for each cell line in this study. Confirmation of a point mutation was possible when the sensitive cell line and its corresponding resistant one showed a lower C_p_ value than the other lines (Table 2). For instance, the sensitive and resistant AU-565 cells exhibited an inferior value for the TP53 and SMAD4 genes (“wt” and “mut” assays). The sensitive and resistant BT-474 cell lines exhibited the lowest difference in their C_p_ value for the NFΚB2 gene for both assays. The EFM-192A and EFM-192A.rTP lines showed the lowest difference in C_p_ values for the NOTCH2 gene. Finally, the SK-BR-3 cell lines displayed the lowest difference in C_p_ values between their “wt” and “mut” trials for the TP53 gene. Thus, qPCR-specific assays confirmed the identity of all the BCCLs that were included in this study based on their mutational profile.

### 2.3. Alterations in the Phosphorylation Pattern of the HER Family Receptors in HER2-Positive BCCLs with Acquired Resistance to Trastuzumab and Pertuzumab

Previous reports have demonstrated that trastuzumab inhibits the activation of downstream signaling pathways by HER2 homodimers, whereas pertuzumab impedes HER2 heterodimerization with EGFR, HER3, and HER4, along with the downstream signaling pathways triggered by these heterodimers [45]. Thus, we evaluated the possible dysregulated activity and expression of HER family members in our HER2-positive breast cancer cell models, which were either sensitive or resistant to dual anti-HER2 therapy.

No significant differences were observed in EGFR levels (total form nor phosphorylated) in the AU-565 and AU-565.rTP cell lines. In both cell lines, reduced levels of pHER3 were observed in the presence of trastuzumab and pertuzumab, particularly in the resistant AU-565.rTP cell line. Of note, this resistant cell line showed a decreased expression of pHER4 under both the control and treatment conditions compared with the sensitive one. Higher levels of pHER2 were identified in the sensitive cell line (Figure 4A). We identified an increased expression of the phosphorylated forms of four HER receptors in the BT-474.rTP cell line compared with its corresponding sensitive line. EGFR, phosphorylated and total form, showed marked differences between the two cell lines. Furthermore, we observed a reduction in the expression of HER2-, HER3-, and HER4-phosphorylated forms in both the sensitive and resistant cell lines upon dual treatment (Figure 4B). In the EFM-192A.rTP cell line, a decrease in all phosphorylated forms was identified, and a significant decrease in pHER3 was observed in both of the cell lines in the treatment scenario. Reduced levels of HER4 expression were observed in the EFM-192A.rTP line compared with its parental line (Figure 4C). Conversely, in the SK-BR-3 and SK-BR-3.rTP cell lines, the levels of EGFR, HER2, and HER3 did not show significant differences. We identified higher levels of HER4 expression in the resistant cell lines, and the phosphorylated form of HER3 showed a marked decrease in the SK-BR-3 line after combined treatment for 24 h. In the presence of the treatment, a smaller reduction in pHER3 levels was observed in the corresponding resistant cell line (Figure 4D).

In summary, our results suggest that the acquisition of resistance to dual anti-HER2 therapy with trastuzumab and pertuzumab may affect the expression and activity of HER family members, particularly HER4. The increased expression of phosphorylated forms of the four HER receptors in the BT-474.rTP cell line highlights the potential role of HER family members in the acquisition of resistance to trastuzumab and pertuzumab in this model.

### 2.4. Acquisition of Trastuzumab and Pertuzumab Resistance Causes Alterations in the Phosphorylation Patterns of the Intracellular Signalling Pathways

The present study investigated the activity of downstream effector pathways, such as PI3K/AKT and MAPK, in sensitive and resistant HER2-positive BCCL models treated with trastuzumab and pertuzumab. We observed higher levels of AKT-phosphorylated forms at Ser473 and Thr308 in the AU-565 and AU-565.rTP cell lines after 24 h than after 6 h (Figure 5A). Following dual treatment, we detected a reduction in the expression of pAKT Ser473 and pAKT Thr308 in both of the cell lines. No significant modifications were found between the conditions compared in the phosphorylated and total forms of ERK and S6. However, a subtle increase was observed in the phosphorylated form of P38 in the resistant cell line in the absence of treatment. In the BT-474 and BT-474.rTP cell lines, we observed no differences in the levels of pAKT Ser473 or Thr308 under the control scenario (Figure 5B). Nevertheless, the levels of pAKT, especially at Thr308, only decreased in the presence of combined treatment in the sensitive cell line. Similar results were observed in the levels of pS6 after 6 h of treatment. In the resistant cell line, a significant increase in pERK and pP38 expression, particularly in the phosphorylated form of P38, was observed compared to the sensitive cell line. No differences in pERK expression were observed in the BT-474.rTP line after treatment with trastuzumab and pertuzumab for 6 h or 24 h. In the EFM-192A and EFM-192A.rTP cell lines, treatment with trastuzumab and pertuzumab caused a marked reduction in pAKT Thr308 expression at 6 h and 24 h (Figure 5C). Conversely, we observed higher expression levels of pERK and pP38 in the EFM-192A.rTP cell line compared to the parental line. Dual treatment caused a reduction in pERK levels at 6 h in the SK-BR-3 line, but this change was not observed in its corresponding resistant cell line. In the SK-BR-3 and SK-BR-3.rTP cell lines, we identified a decrease in both pAKT forms following the treatment with anti-HER2 therapy (Figure 5D). Additionally, the resistant cell line showed a smaller increase in pERK in the absence and presence of combined treatment compared to the sensitive cell line. We did not observe differences in either cell line regarding the levels of S6 and P38 phosphorylation. However, the total form of S6 exhibited higher levels in the resistant cell line. In summary, no major changes in the phosphorylation levels of the two forms of AKT were observed in any of the resistant lines compared to the sensitive ones. In fact, subsequent treatment with the combination of both drugs caused a decrease in pAKT levels in all the models, as would be expected under basal conditions when dual treatment blocks the activation of the PI3K/AKT pathway. However, a significant increase in pERK, and especially pP38 levels, was observed in the resistant lines, suggesting that the MAPK pathway may be involved in the acquisition of resistance to trastuzumab and pertuzumab.

### 2.5. Trastuzumab and Pertuzumab Resistance Acquisition Modifies Metabolic, Mitochondrial, or Ribosomal Cellular Processes

Compared to luminal cell lines, HER2-positive cells that lack the expression of hormonal receptors exhibit increased aggressiveness and better responses to specific drugs. Previous studies have demonstrated that Pearson’s correlation test indicates a considerable relationship between the molecular signature and the biological response to trastuzumab in HER2-positive cells, making it an outstanding design to study anti-HER2 treatment responses [46,47]. Thus, we decided to perform a proteomic analysis on the SK-BR-3 cell line and its corresponding resistant line due to the cellular and molecular differences observed between them and due to its molecular classification as a HER2-positive cell line. For our study, we performed a LC-MS/MS analysis to investigate differences between the sensitive and resistant SK-BR-3 cell lines under basal conditions, in the absence of anti-HER2 treatment. We identified 4239 proteins, 4110 of which were able to be quantified. A total of 618 proteins were differentially expressed with a *p*-value < 0.05, 349 of which were upregulated in the resistant cell line compared to the sensitive cell line, while 269 proteins were downregulated.

When we applied robustness criteria in the analysis of peptide spectra identifications and quantifications (Mascot score > 70, at least one unique peptide, a q-value < 0.05, an abundance ratio variability < 30%, and an abundance rate > 1.5), both lists became more restricted. Specifically, we found 83 more abundant proteins and 118 less abundant proteins in the resistant strain. The top proteins in each list are presented in Table 3 (overexpressed) and Table 4 (downregulated). The most abundant proteins in the resistant line were associated with processes like ribosome formation, mitochondrial activity, or metabolism as an oxidative response to stress, which align with the characteristics of more proliferative and resistant cells. On the other hand, the downregulated proteins in Table 4 mainly include small proteins related to metabolism, oxidoreductase activity, amino acid degradation, and nucleotide metabolism. Additionally, peroxisome proteins, ribonucleoproteins of the 7SK snRNP complex, cell–substrate junction proteins, and actin capping proteins were also diminished in the resistant cells. These findings support the cellular observations of increased cell proliferation, drug resistance, and invasive capacity in the resistant cells.

## 3. Discussion

One of the main clinical challenges in cancer treatment is the emergence of resistance to therapy, both in terms of risk to patient survival and the economic cost of using alternative therapies. Thus, the development of laboratory models of acquired resistance represents an important tool to investigate the mechanisms underlying therapeutic resistance and to suggest new therapeutic approaches to treat resistant tumors. We generated HER2-positive breast cancer cell models with resistance to trastuzumab and pertuzumab via prolonged exposure to both antibodies in a panel of four HER2-positive BCCLs and characterized them in terms of their proliferative and clonogenic potential. We have also assessed these new cell lines for the expression of receptors defined during clinical diagnosis via immunohistochemistry and authenticated them through mutational analyses. We compared the expression and phosphorylation status of HER family receptors and the PI3K/AKT and MAPK pathways between the sensitive and resistant cell lines and after treatment with dual therapy. Finally, we performed a proteomic study in one of the BCCLs based on a label-free mass spectrometry analysis and subsequent functional interpretation in terms of differential protein expression.

The generation of resistant cell lines through the method of continuous low-dose drug exposure is a time-consuming but effective task [36]. In our case, it took about a year to generate BCCLs resistant to dual anti-HER2 therapy, but their resistance rates in terms of proliferation were high. In addition, we were able to subclone cell populations with a more refined degree of resistance in almost all cases. Furthermore, clonogenic assays allowed for resistance evaluation over longer periods of time and confirmed resistance acquisition in all models. No morphological changes were observed in any of the cell lines during the acquisition of resistance. Proteomic analysis showed that certain cellular structures were impaired in their maintenance; however, these molecular changes did not translate into visible morphological changes. At the same time, we subjected all the parental and corresponding resistant cell lines to the mutation-specific authentication panel described in the Cancer Cell Line Encyclopedia. The mutational analysis confirmed the identity of all the cell lines that were used in this project and ruled out cross-contaminations [48]. Similarly, to confirm the absence of changes in the phenotype characterizing the molecular subtype of each cell line after the acquisition of resistance, we assessed hormone receptors and HER2 via IHC [49]. Thus, we validated the molecular profile of all the cell lines and confirmed their corresponding subtype: HER2-enriched luminal or true HER2-positive.

The HER family of receptors acts as critical regulators of normal cellular processes, but it has also become apparent that their dysregulation leads to tumor development. In breast cancer, the HER2 receptor is often deregulated, and tumors with HER2 amplification or overexpression have a poorer prognosis and clinical course [50,51]. The use of specific anti-HER2 therapies, especially dual therapy with pertuzumab and trastuzumab, has been demonstrated to be effective, achieving a more complete signaling inhibition [52]. However, due to the blockade of signaling mediated by HER2 and HER3, the appearance of compensatory mechanisms mediated by other members of the HER family has been described [53]. Therefore, we assessed changes in the phosphorylation and expression of HER family receptor members in our resistant models. We found no major changes. The relationship between variations in HER2 receptor abundance levels and the development of resistance to monoclonal antibody anti-HER2 therapy has been a controversial topic since the approval of trastuzumab as a treatment several decades ago. Conflicting reports exist regarding the levels of the HER2 receptor in cancerous cells. The levels of receptors may either decrease, increase, or remain unchanged, depending on the type of cancer, cell line, presence or absence of mutations in the HER2 gene, and experimental conditions. It is unclear as to whether trastuzumab downregulates HER2 expression, as several studies have demonstrated no change in its receptor levels after trastuzumab treatment [54,55]. However, other studies have reported that trastuzumab may cause the internalization and degradation of HER2, leading to reduced receptor signaling. In several cases, tumor cells may respond to anti-HER2 therapy by upregulating HER2 expression due to selective pressures. This can occur through increased HER2 gene transcription or changes in the post-transcriptional and post-translational processes that affect HER2 protein levels.

Several mechanisms can contribute to treatment resistance in the context of acquired resistance to anti-HER2 therapy, and variations in HER2 protein levels may play a role. Resistance to anti-HER2 therapies is often multifactorial, and changes in HER2 expression are just one aspect of the complex landscape of resistance mechanisms. Our study found that the development of resistance to anti-HER2 therapies in our cell lines did not result in a significant change in the levels of the receptor protein. The number of HER2 signals detected in the cell lines that acquired resistance was similar to the signals detected in their corresponding parental cell line. In a previous publication, we reported on the development of resistance to trastuzumab in HER2-positive breast cancer cell lines [35]. This study confirmed that matched sensitive and resistant populations did not show changes in their molecular profiles of the markers, ruling out the acquisition of secondary resistance to antibody-based therapy through substantial changes at the HER2 receptor expression level. Although there are contradictory reports in the literature, we have confirmed that trastuzumab does not downregulate HER2 receptors in acquired-resistant breast cancer cell lines even after months of treatment. However, it is essential to monitor the molecular profile of tumors, including their HER2 status and expression levels, during the course of treatment to understand and address their acquired resistance.

Interestingly, in the resistant lines AU-565, EFM-192A, and SK-BR-3, we observed a reduction in pHER3 levels following the treatment with dual therapy, suggesting that the acquisition of a resistant phenotype is not due to increased activation of this receptor. It is therefore conceivable that HER2 interacts with other tyrosine kinase receptors, including EGFR or HER4, to cause the development of resistance. In our study, we identified a marked overexpression of EGFR in the BT-474.rTP cell line after its acquisition of resistance. EGFR overexpression has been observed in 15–30% of breast carcinomas and has been associated with large tumors and poor clinical outcomes. In particular, EGFR has been frequently overexpressed and associated with a poor prognosis in triple-negative breast cancer [56,57]. EGFR/HER2 heterodimers have been identified to increase the metastatic potential of BCCLs [58], and simultaneous over-phosphorylation between EGFR and HER2 has been described in samples from patients with metastatic breast cancer [59]. Increased EGFR expression has also been previously described in trastuzumab-resistant cell lines compared to sensitive ones. Kwon et al. described an interaction between cytoplasmic ERα and EGFR/HER2 heterodimers in the development of trastuzumab resistance [60]. Further research is needed to determine a similar role of EGFR overexpression and overactivation through its phosphorylation state in the development of resistance to dual anti-HER2 therapy.

On the other hand, the role of HER4 in breast cancer is controversial. Regarding anti-HER2 antibody therapy, several studies have shown that the overexpression of nuclear HER4 mediates trastuzumab resistance in HER2-positive breast cancer. In addition, the HER4–YAP1 axis has been reported to promote trastuzumab resistance in HER2-positive gastric cancer [61,62]. In the BT-474.rTP and SK-BR-3.rTP cell lines, we identified increased HER4 expression, suggesting a possible activation of alternative signaling pathways following the acquisition of resistance. Further research is needed to determine a possible mechanism of resistance to the trastuzumab and pertuzumab combination induced via increased HER4 expression.

In summary, our results suggest that the acquisition of resistance to dual anti-HER2 therapy with trastuzumab and pertuzumab may affect the expression and activity of HER family members, particularly HER4. The increased expression of the phosphorylated forms of the four HER receptors in the BT-474.rTP cell line highlights the potential role of the HER family members in the acquisition of resistance to trastuzumab and pertuzumab. The observed reduction in the expression of the phosphorylated forms of HER2, HER3, and HER4 in response to this treatment indicates that T + P therapy can effectively reduce the activity of these receptors. These findings provide insights into the mechanisms underlying resistance to dual anti-HER2 therapy and may help in the development of new therapeutic strategies.

A wealth of literature has been published on the key role of the PI3K/AKT and MAPK pathways in the pathophysiology of breast cancer and in the development of resistance to targeted therapies [63,64,65]. In particular, they play a crucial role due to their activation by HER2 homodimers and HER2–HER3 heterodimers. Therefore, we aimed to explore whether the acquisition of resistance to anti-HER2 therapies in HER2-positive BCCLs would occur through changes in the expression and/or phosphorylation in the PI3K/AKT and MAPK pathways. Our resistant cell lines, in particular the AU-565.rTP, BT-474.rTP, and EFM-192A.rTP cell lines, showed a marked upregulation of both forms of pAKT Ser473 and Thr308. Subtle changes in S6 and its phosphorylated form were identified. These results suggest an activation of AKT following the acquisition of resistance to dual anti-HER2 therapy, which has previously been associated with trastuzumab resistance [66,67].

For the MAPK pathway, we evaluated ERK 1/2 (ERK) and P38 activation based on previous studies reporting the role of these signaling nodes in HER2-positive breast cancer and drug resistance [68,69,70]. Interestingly, we identified common variations in pERK levels in all the resistant cell lines (except AU-565.rTP), suggesting their association with resistance mechanisms. These data are consistent with our previous work on ERK activation by CCL5 in acquired trastuzumab resistance [71]. Recent studies have uncovered the activation of MEK/ERK signaling in tumors with acquired anti-HER2 therapy [72,73]. Regarding P38, we identified increased expression and phosphorylation in the resistant cell lines BT-474.rTP and EFM-192A.rTP, especially in the former. Previous studies have described a key role of P38 signaling in drug resistance in the context of breast cancer, particularly to trastuzumab, letrozole, or chemotherapy [74,75,76]. The overactivation of the ERK- and P38-mediated signaling pathways in these cell lines suggests a major role in the mechanisms of acquisition of resistance to trastuzumab and pertuzumab.

Proteomic analyses can provide biological insights that can guide therapeutic strategies and ultimately improve treatment efficacy and patient responses. Based on the proteomic analysis performed in the SK-BR-3 cell line and its corresponding resistant line, we identified changes in the abundance levels of several proteins, as shown in Table 3 and Table 4, presumably due to the acquisition of resistance. Taken together, most of these changes in abundance levels point towards dysregulation in processes related to cell growth and proliferation, as well as to the maintenance of cellular structures and drug resistance. Among the processes associated with overexpressed proteins, we identified ribosome formation, mitochondrial activity, and changes in metabolism related to oxidative stress. One possible explanation is that increased proliferation of tumor cells, particularly in resistant cell lines, requires increased protein translation and synthesis, and thus increased ribosome formation. Several studies have shown that proteins involved in ribosome biogenesis play a key role in radioresistance and chemoresistance in various cancer types [77,78,79], and in particular, the involvement of RPS6 in the resistance to trastuzumab and lapatinib has been demonstrated in gastric cancer models [80]. On the other hand, alterations in mitochondrial activity and metabolic profiles have been associated with the development of resistance to various therapies in breast cancer. Thus, several studies have reported the existence of alterations in mitochondrial function as well as alterations in the metabolism of cells that exhibit resistance to tamoxifen [81,82]. Regarding the development of trastuzumab resistance, several metabolic alterations have been identified: increased glycolysis by HSF1 and LDH-A contributes to the development of trastuzumab resistance [22]; activation of t-DARPP and IGF-1R in the stimulation of glycolysis confers resistance to trastuzumab [83]; and the development of resistance to anti-HER2 therapies has also been associated with the reprogramming of lipid metabolism [84].

In addition, our proteomic findings also revealed reduced expression levels of HEXIM1 and MEPCE in the resistant cell line, which are two components of the 7SK snRNP complex responsible for facilitating the release of the paused RNA polymerase II complex [85]. HEXIM1 has been identified as a tumor suppressor, and reducing its level of expression has appeared to be associated with resistance to tamoxifen in breast tumors [86], anti-androgens in prostate cancer [87], and increased progression and decreased therapeutic sensitivity in triple-negative breast cancer [88]. We also identified the downregulation of several actin-capping associated proteins in the resistant cell line. The role of these proteins in cancer has remained unclear due to conflicting results: while some studies have classified them as oncogenes, others have proposed that they predominantly encompass tumor suppressor functions [89,90]. In addition, only a limited number of studies have investigated the potential role of these proteins in the development of resistance in cancer, particularly in ovarian cancer and gliomas [91,92]. Therefore, it would be necessary to investigate the potential role of all these under-expressed proteins and their various complexes in the development of resistance to anti-HER2 therapies.

The changes in these proteins observed in our study suggest that they may contribute to the acquisition of resistance to trastuzumab and pertuzumab in HER2-positive breast cancer that does not express the hormone receptor. To our knowledge, the potential contribution of these proteins to resistance to dual HER2 blockade with trastuzumab and pertuzumab has not been investigated. Functional and clinical validations will undoubtedly be required to identify their potential role in the molecular mechanisms associated with anti-HER2 resistance. In any case, a larger study will be essential to confirm the potential role of these proteins in the acquisition of resistance to therapy in this subtype of breast cancer.

## 4. Materials and Methods

### 4.1. Cell Cultures and Reagents

The human breast cancer cell lines (BCCLs) AU-565 (CRL-2351), BT-474 (HTB-20), and SK-BR-3 (HTB-30) were purchased from the American Type Culture Collection, and EFM-192A (ACC-258) was obtained from the German Tissue Repository DSMZ. BT-474 and SK-BR-3 cells were maintained in DMEM/F-12 (Sigma-Aldrich, Steinheim, Germany) supplemented with 10% heat-inactivated fetal bovine serum (Gibco, Thermo Fisher Scientific, Waltham, MA, USA), 2 mmol/L glutamine (GlutaMAX, Gibco), and 1% penicillin G–streptomycin (P/S, Gibco). AU-565 and EFM-192A cells were cultured in RPMI 1640 (Gibco) supplemented with 10% and 20% heat-inactivated FBS, respectively, 2 mmol/L glutamine, and 1% P/S. The cell lines were grown at 37 °C under a humidified atmosphere with 5% CO_2_. All the cell lines that were used for these experiments were free of mycoplasma contamination, as assessed using a previously described method [35]. The recombinant humanized monoclonal anti-HER2 antibodies trastuzumab (Herceptin^®^, Genentech Inc., South San Francisco, CA, USA) and pertuzumab (Perjeta^®^, Genentech Inc.) were kindly provided by the Fundación Jiménez Díaz Hospital pharmacy.

### 4.2. Establishment of Trastuzumab and Pertuzumab-Resistant BCCLs

Trastuzumab and pertuzumab-resistant cell lines were established through continuous exposure to trastuzumab and pertuzumab, as previously described [35,93]. The cells were treated with both treatments for a period of 8 months for AU-565, EFM-192A, and SK-BR-3 and 15 months for BT-474. The algorithm described by O’Brien et al. estimates the correlation between the growth rates of treated and untreated cells based on their cell doubling times. Resistance rates based on this procedure were determined monthly through cell proliferation assays. Cells with a fold growth ≥ 1.20 were deemed sensitive to this treatment [44]. Following resistance establishment, the cells were cultured with a trastuzumab and pertuzumab maintenance dose of 15 g/mL and 20 g/mL, respectively. To serve as procedural controls, the parental lines were grown without treatment in parallel to maintain their sensitivity to these drugs.

### 4.3. Cell Proliferation Assays

Sensitive and resistant cells were seeded in triplicate in 12-well plates at a density of 2.5 × 10^4^ cells for AU-565 and SK-BR-3 and 5 × 10^4^ cells for BT-474 and EFM-192A, and were allowed to adhere and enter the growth phase before being treated with vehicle, 15 µg/mL trastuzumab, 20 µg/mL pertuzumab, or the combined treatment for 7 days in the appropriate culture medium. The culture medium and treatments were replaced every 3 days. The cells were then harvested via trypsinization and counted with trypan blue using the TC20 automated cell counter (BioRad, Hercules, CA, USA). All experiments were repeated three times in triplicate for each concentration.

### 4.4. Clonogenic Assays

Sensitive and resistant cells were seeded in T-25 flasks at a density of 1 × 10^3^ cells for AU-565 and SK-BR-3 and 2 × 10^3^ cells for BT-474 and EFM-192A, and were allowed to adhere and enter the growth phase before being treated with or without 1 µg/mL trastuzumab and 1.3 µg/mL pertuzumab for 21 days in the appropriate culture medium. The appropriate culture medium and treatments were replaced every 3 days. After 21 days, the colonies were stained with 1% crystal violet dye, and colony number and area were estimated using the ImageJ program (NIH). Three independent experiments were performed for each condition and cell line.

### 4.5. Immunohistochemistry

Cell pellets were generated from 0.5 × 10^7^ to 1 × 10^7^ cells and included in FFPE blocks. Immunostaining was performed using 3-μm FFPE sections of breast cancer cellular pellets placed on plus-charged glass slides on an Omnis platform for estrogen and progesterone receptors and an Autostainer link 48 platform for HER2 (Agilent Technologies, La Jolla, CA, USA). After deparaffinization, heat antigen retrieval was performed in a pH-9 EDTA-based buffered solution (Agilent). Endogenous peroxidase was quenched. The following primary antibodies were incubated for 30 min at RT: anti-estrogen receptor α (clone EP1) rabbit monoclonal antibody (GA08461-2, Agilent, ready to use), anti-progesterone receptor (clone PgR 1294) mouse monoclonal antibody (GA090, Agilent, ready to use), and HercepTest (SK00121-2, Agilent). Antigen–antibody reactions were detected via incubation with an anti-mouse/rabbit Ig–dextran polymer coupled with peroxidase (GV800, Agilent). The sections were then visualized with 3,3′-diaminobenzidine and counterstained with hematoxylin. All immunohistochemical staining was performed on a Dako Autostainer platform.

### 4.6. DNA Extraction

DNA was extracted from cellular pellets using the High Pure PCR Template Preparation kit (Roche, Mannheim, Germany), following the manufacturers’ instructions. The extracts were quantified in a NanoDrop 2000 spectrophotometer (Thermo Fisher Scientific) at 260 nm and subsequently stored at −20 °C.

### 4.7. Authentication Profiling of BCCLs by Mutational Analysis

As previously reported by our group [35], and according to the panel of mutations described in the Cancer Cell Line Encyclopedia (CCLE) (http://www.broadinstitute.org/ ccle, accessed on 6 June 2013), specific point mutations were chosen for different genes in the different cell lines, so that every cell line was unequivocally identified via a specific mutation. Briefly, to establish the mutational profile of each BCCL, one PCR was prepared to evaluate the wild-type (wt) profile and one for the mutated (mut) profile. The qPCR conditions consisted of an initial denaturation cycle for 10 min at 95 °C, followed by 45 cycles in two/three steps (one of 10 s at 95 °C, a second of 30 s at 60–64 °C, and a conditional step of 20 s at 72 °C), and, finally, an unlimited cycle of cooling at 4 °C. The results that were obtained and the crossing point (Cp) values were processed using LightCycler 480 v1.5.0 software (Roche) and were calculated based on the second derivative method.

### 4.8. Protein Extraction and Quantification

The cells were seeded in 6-well plates at a density of 5 × 10^5^ (AU-565 and SK-BR-3) or 1× 10^6^ (BT-474 and EFM-192A) cells per well and were allowed to adhere for 24 h. Then, the cells were treated with vehicle or the combined treatment with 15 µg/mL trastuzumab and 20 µg/mL pertuzumab for 6 h or 24 h. Subsequently, the cells were washed with 3 mL of PBS at RT. Next, the cells were scraped in the presence of a 150-µL lysis buffer (RIPA buffer, peptidase inhibitors, and phosphatase inhibitors) at 4 °C and transferred to a 1.5 mL tube. The cells were incubated in the lysis buffer for 20 min at 4 °C and sonicated afterwards. Then, the cell lyzates were spun at 13,000× *g* for 10 min at 4 °C, and the supernatant was retained and stored. The protein extracts were quantified using the Pierce BCA protein assay kit (Thermo Fisher Scientific), following the manufacturer’s instructions.

### 4.9. Western Blotting (WB) Analysis

Protein aliquots were prepared at 1 µg/µL in 4× Laemmli loading buffer and boiled at 95 °C for 15 min. Twenty µL of protein extract was loaded in a 10% polyacrylamide gel (SDS–PAGE). Next, proteins were transferred to a nitrocellulose membrane for 90 min at 130 V and 4 °C. The membrane was blocked (5% BSA in PBST 1×) for 1 h and then incubated with the primary antibodies at 4 °C overnight under agitation. The concentrations used were as follows: HER3 (1:500; Thermo Fisher Scientific), pHER3 Tyr1197 (1:500), HER3 (1:500), pHER2 Tyr1221/1222 (1:1000), HER2 (1:1000), pEGFR Tyr1173 (1:500), EGFR (1:500), pHER4 Tyr1284 (1:500), HER4 (1:500), pAKT Thr308 (1:1000), pAKT Ser473 (1:1000), AKT (1:1000), pp44/42 MAPK (ERK1/2) Thr202/Tyr204 (1:1000), p44/42 MAPK (ERK1/2) (1:1000), pS6 ribosomal protein (pS6) Ser235/236 (1:1000), S6 ribosomal protein (S6) (1:1000); pP38 MAPK Thr180/Tyr182 (pP38) (1:1000), P38 MAPK (P38) (1:1000), (Cell Signaling, Danvers, MA, USA), and β-actin (1:5000; Sigma-Aldrich, St. Louis, MO, USA). All the primary antibodies were rabbit in origin, except for the anti-HER3 antibody, which was mouse. Then, the membranes were washed 3 × 10 min in 1× PBST and incubated with a secondary antibody (diluted in 2.5% BSA in 1× PBS) at RT for 1 h. ECL-anti-mouse and ECL-anti-rabbit secondary antibodies attached to peroxidase (HRP; GE Healthcare, Chicago, IL, USA) were used at a concentration of 1:5000. The membranes were washed 3 × 10 min again and immersed in the detection reagent (Immobilon Crescendo Western HRP substrate, Merck Millipore, Burlington, MA, USA) for 1 min. Densitometry and quantification of proteins were carried out using ImageJ v1.54d software.

### 4.10. Mass Spectrometry Analysis

A total of 2.5 × 10^6^ cells of the SK-BR3 and SK-BR3.rTP cell lines were seeded in a P100 plate and were allowed to adhere for 24 h in complete medium. Then, the cells were washed with cold PBS, and lyzates were obtained by scraping the cells in RIPA lysis buffer supplemented with phosphatase and protease inhibitors. Protein concentration was determined using the BCA protein assay kit. The proteomic analysis was performed in the Proteomics Unit of the Complutense University of Madrid. Cell lyzate triplicates of 100 µg were precipitated with methanol/chloroform. The pellet, which contained the proteins, was resuspended in 20 μL of 8 M urea for digestion. The precipitated proteins were reduced with DTT at 56 °C for 1 h and then alkylated with 25 mM iodoacetamide for 1 h at RT. The lyzates were incubated with a ratio of 1 μg of recombinant trypsin overnight at 37 °C. Desalination and concentration were performed with Poros R2 and quantified via fluorimetry (Qubit). They were dried via vacuum centrifugation (SpeedVac, Savant, Thermo Fisher) and were reconstituted to each have a concentration of 0.2 µg/µL of 2% ACN and 0.1% formic acid and stored at −20 °C until analysis. The peptides (1 μg) were analyzed via liquid nanochromatography (nano Easy-nLC 1000, Thermo Scientific, Bremen, Germany) coupled to a Q-Exactive HF high-resolution mass spectrometer (Thermo Scientific). The peptides were concentrated on-line via reversed-phase chromatography using an Acclaim PepMap 100 guard column (Thermo Scientific) and were then separated on a Picofrit C18 reversed-phase analytical column (Thermo Scientific). MS/MS data were acquired in the data-dependent acquisition (DDA) mode of the MS.

### 4.11. Protein Identification and Quantification

MS/MS spectra acquired on the samples were analyzed using Proteome Discoverer v2.5 software (Thermo Scientific) with the MASCOT v.2.8 search engine. The UniProt database with taxonomic restriction to humans was used (release 2023_03). To determine the abundance of the identified peptides and proteins in different isolates, a label-free experiment based on the intensity of the precursor signal was performed. The percolator algorithm was used to estimate the false discovery rate (FDR) and filtered using a q-value < 0.01 for proteins identified with high confidence. The number of proteins, peptides, and identified spectra in the database used was summarized in a final report.

### 4.12. Protein Data Analysis

The Proteome Discoverer application includes a statistical feature (ANOVA background) for assessing the significance of differential expression by providing *p*-values and adjusted *p*-values (q-values) for those ratios. Once non-specific proteins, highly abundant in all cells, were discarded, only proteins identified with high confidence (FDR < 1%) with at least one unique peptide, an abundance ratio variability < 30%, a q-value < 0.05, and a fold change > 1.5 were considered to be differentially expressed between groups. The MS proteomics data have been deposited to the ProteomeXchange Consortium via the PRIDE partner repository with the dataset identifier PXD045804. Proteome Discoverer also includes a principal component analysis (PCA) to identify the major components in a protein dataset using abundance-normalized values. The input data used for the PCA were the master proteins identified with high confidence in the database, without taking contaminating proteins into account.

### 4.13. Statistical Analysis

All the measured data were expressed as the means ± standard deviations of at least three replicates (unless otherwise indicated). Statistical significance was analyzed through two-sided, unpaired *t*-tests using GraphPad Prism 8.0.1 software (GraphPad Software, La Jolla, CA, USA) (*: *p*-value < 0.05, **: *p*-value < 0.01, and ***: *p*-value < 0.001). This work was performed in accordance with the Reporting Recommendations for Tumor Marker Prognostic Studies (REMARK) guidelines [94].

## 5. Conclusions

In this study, we produced four novel BCCL-derived cell lines through prolonged exposure to standard anti-HER2 therapy, specifically trastuzumab and pertuzumab. We were able to confirm the identities of the newly established populations by analyzing their molecular profile through mutational analysis via PCR and examining the expression of ER, PR, and HER2 receptors using IHC. After analyzing the cell population post-drug treatment, we affirmed that the derived lines were resistant to trastuzumab and pertuzumab combination and established their resistance rate. The WB biochemical study helped us identify the main molecular changes typically seen in breast cancer. This led us to conclude that the resistance mechanisms produced in BCCLs following the trastuzumab and pertuzumab treatment may be linked to the MAPK pathway. Finally, a proteomic analysis confirmed a significant alteration in the abundance patterns of over 600 proteins, suggesting implications for essential biological processes such as ribosome formation, mitochondrial activity, and metabolism. These changes may play a role in the development of resistance in HER2-positive breast cancer. These experimental breast cancer models, which have developed resistance to trastuzumab and pertuzumab, are presently being utilized in our lab to explore the intricate molecular changes that occur during resistance development. Our main objective was to deliver valuable clinical insights to enhance the effectiveness of therapeutic interventions in the future.

## Figures and Tables

**Figure 1 ijms-25-00207-f001:**
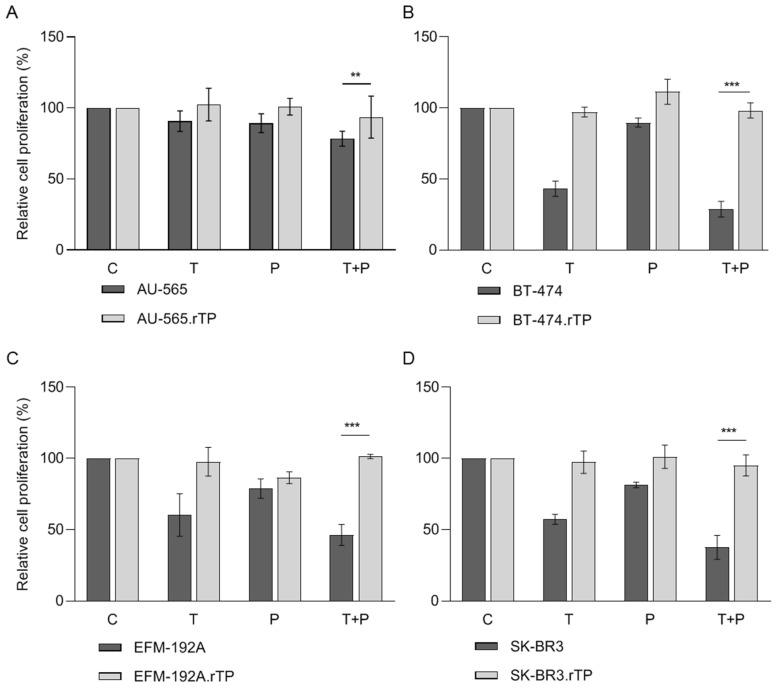
Generation of cell lines resistant to dual anti-HER2 therapy, trastuzumab and pertuzumab. Effect of treatment with 15 µg/mL trastuzumab (T), 20 µg/mL pertuzumab (P), or a combination of T + P for 7 days in a panel of sensitive and acquired-resistant cell lines, after a period of several months of generation of acquired resistance. Viable cells were counted via trypan blue exclusion. Viability was presented as a percentage of the respective DMSO-treated control group (**C**). (**A**) AU-565-sensitive (AU-565) and trastuzumab and pertuzumab-resistant (AU-565.rTP) cells. (**B**) BT-474-sensitive (BT-474) and -resistant (BT-474.rTP) cells. (**C**) EFM-192A-sensitive (EFM-192A) and -resistant (EFM-192A.rTP) cells. (**D**) SK-BR-3-sensitive (SK-BR-3) and -resistant (SK-BR-3.rTP) cells. Error bars represent the standard deviation between replicates (*n* ≥ 3). ** denotes *p* ≤ 0.01; and *** denotes *p* ≤ 0.001.

**Figure 2 ijms-25-00207-f002:**
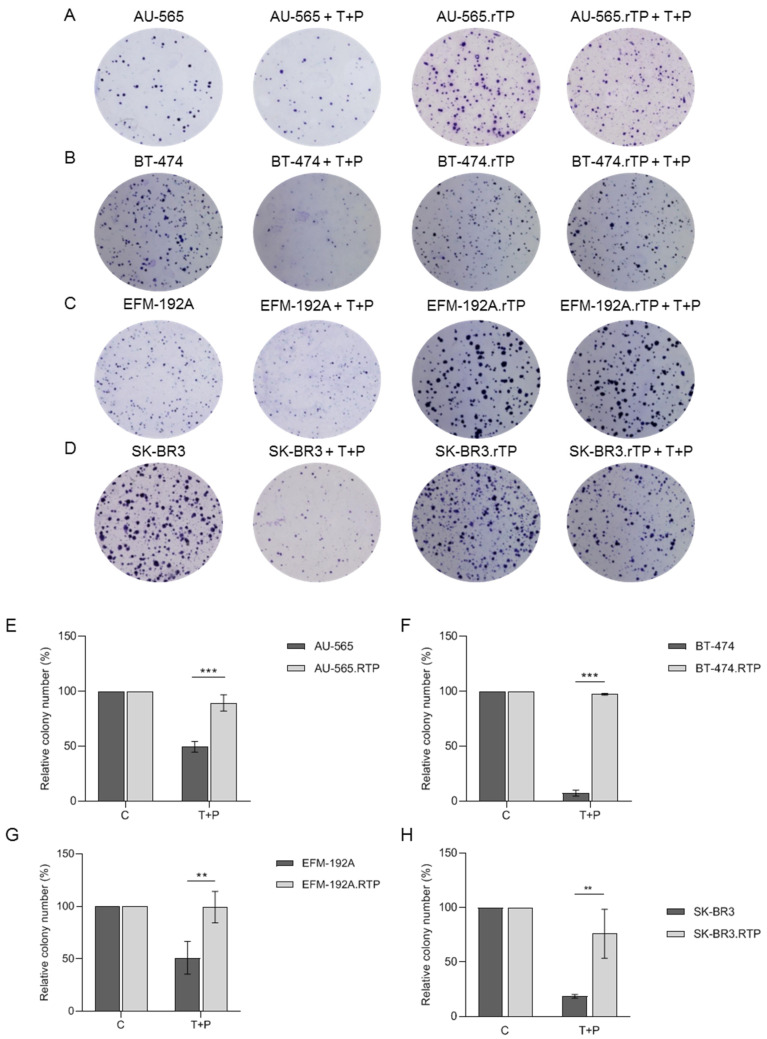
Effect of long-term treatment with trastuzumab and pertuzumab in sensitive and resistant cell lines. Clonogenic capacity of sensitive vs. resistant cell lines was compared after 21 days of treatment with T + P. (**A**–**D**). Representative images of colony formation after staining with 1% crystal violet. (**A**) AU-565 and AU-565.rTP. (**B**) BT-474 and BT-474.rTP. (**C**) EFM-192A and EFM-192A.rTP. (**D**) SK-BR-3 and SK-BR-3.rTP. (**E**–**H**) Relative number of colonies presented as a percentage of the DMSO-treated control colony number. (**E**) AU-565 and AU-565.rTP. (**F**) BT-474 and BT-474.rTP. (**G**) EFM-192A and EFM-192A.rTP. (**H**) SK-BR-3 and SK-BR-3.rTP. Error bars indicate the standard deviation between replicates (*n* ≥ 3). ** denotes *p* ≤ 0.01; *** denotes *p* ≤ 0.001.

**Figure 3 ijms-25-00207-f003:**
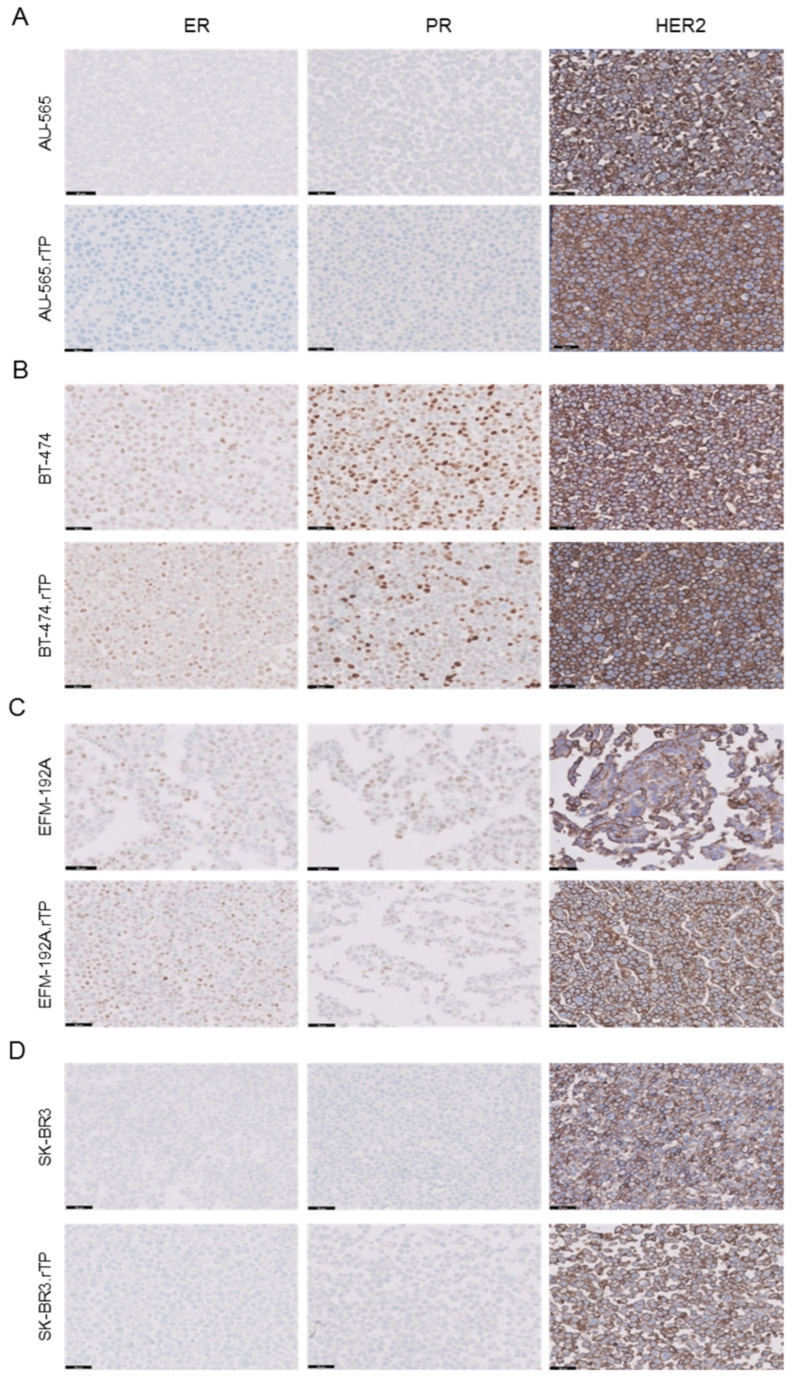
Characterization of sensitive and resistant cell lines according to their IHC profiles of ER, PR, and HER2 expression. The cells were prepared as FFPE pellets. Resistant populations showed no change in their molecular profile compared to their sensitive counterparts. (**A**) AU-565 and AU-565.rTP. (**B**) BT-474 and BT-474.rTP. (**C**) EFM-192A and EFM-192A.rTP. (**D**) SK-BR-3 and SK-BR-3.rTP. BT-474- and EFM-192A-sensitive and resistant cell lines were confirmed as luminal (ER positive, PR positive, and HER2 positive). AU-565- and SK-BR-3-sensitive and resistant lines were determined to be HER2 positive and hormonal receptor-negative. The black line represents a length of 50 µm. Magnification: ×200.

**Figure 4 ijms-25-00207-f004:**
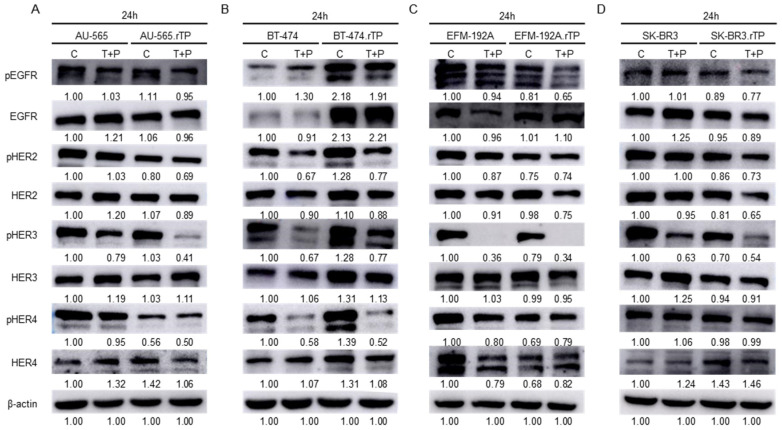
Evaluation of the phosphorylation pattern of HER family receptors via WB. Sensitive and resistant HER2-positive BCCLs were treated with T + P for 24 h. (**A**) AU-565 and AU-565.rTP. (**B**) BT-474 and BT-474.rTP. (**C**) EFM-192A and EFM-192A.rTP. (**D**) SK-BR-3 and SK-BR-3.rTP. Representative images of three replicates are depicted. Relative abundance levels of proteins were determined via densitometric analyses of the images.

**Figure 5 ijms-25-00207-f005:**
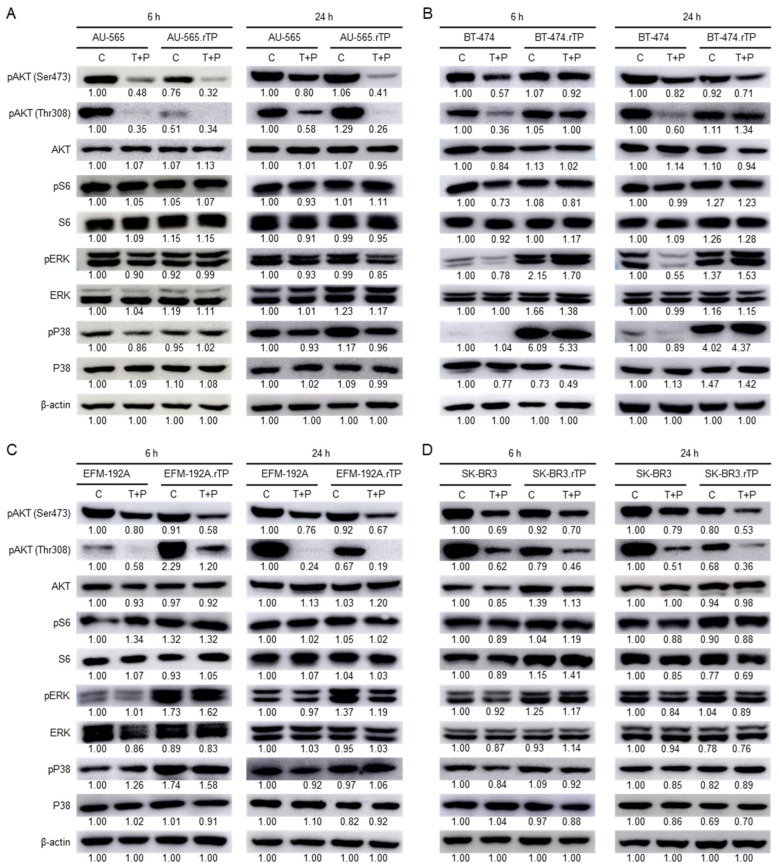
Analysis of the protein phosphorylation patterns of the PI3K/AKT and MAPK pathways was assessed via WB in the sensitive and resistant cell lines. In the eight models, AKT and S6 (the PI3K/AKT pathway), as well as the ERK and P38 (the MAPK pathway) proteins, were examined under control conditions and with T + P treatment for 6 h or 24 h. (**A**) AU-565 and AU-565.rTP. (**B**) BT-474 and BT-474.rTP. (**C**) EFM-192A and EFM-192A.rTP. (**D**) SK-BR-3 and SK-BR-3.rTP. Representative images of three replicates are depicted. The relative abundance levels of these proteins were determined via densitometric analyses of these images.

**Table 1 ijms-25-00207-t001:** Response of parental and derived HER2-positive BCCLs to combined trastuzumab and pertuzumab treatment. Cells were classified as sensitive (S) or resistant (R) based on their proliferation capacity response to a combination of 15 µg/mL trastuzumab and 20 µg/mL pertuzumab for 7 days. Response to each drug was quantified by calculating the fold change in the growth rate (∆GR) of the treated cells relative to the non-treated cells. Resistance was defined as a ∆GR < 1.20 in all cases.

Cell Line	Cell Viability (T + P)	Growth Rate (Fold Change) (∆GR)	Resistance
AU-565	75%	1.20	S
AU-565.rTP	106%	0.97	R
BT-474	29%	3.58	S
BT-474.rTP	85%	1.10	R
EFM-192A	55%	1.47	S
EFM-192A.rTP	87%	1.00	R
SK-BR3	39%	1.67	S
SK-BR3.rTP	95%	1.02	R

**Table 2 ijms-25-00207-t002:** C_p_ differential for amplification curves between the “wt” and “mut” assays for each HER2-positive BCCL.

BCCL	Mutation											
	NFΚB2			TP53			SMAD4			NOTCH2		
	wt	mut	∆	wt	mut	∆	wt	mut	∆	wt	mut	∆
AU-565	34.26	40.00	5.74	35.32	28.61	−6.71	34.31	34.37	−0.04	40.00	40.00	0.00
AU-565.rTP	33.13	40.00	6.87	38.15	37.96	−10.18	32.86	35.47	2.61	40.00	40.00	0.00
BT-474	35.04	34.66	−0.38	31.69	40.00	8.31	33.10	40.00	6.90	40.00	40.00	0.00
BT-474.rTP	34.56	33.99	−0.57	29.98	40.00	10.02	29.92	40.00	10.08	40.00	40.00	0.00
EFM-192A	31.92	40.00	8.08	31.06	40.00	8.94	31.46	40.00	8.54	39.25	35.26	−3.99
EFM-192A.rTP	32.84	40.00	7.16	31.68	40.00	8.32	33.79	40.00	6.21	38.73	36.32	−2.41
SK-BR3	33.95	40.00	6.05	40.00	30.75	−9.25	35.89	40.00	4.11	40.00	40.00	0.00
SK-BR3.rTP	34.25	40.00	5.75	39.65	29.60	−10.05	33.87	40.00	6.13	40.00	40.00	0.00

**Table 3 ijms-25-00207-t003:** List of the top proteins identified via label-free MS/MS as the most abundant in the SK-BR-3.rTP cell line (Mascot score > 70, at least one unique peptide, a q-value < 0.05, an abundance ratio variability < 30%, and an abundance ratio > 2.5). An abundance ratio = 100 means that none of the peptides identified in the resistant cell samples were found in their sensitive parental cells.

UniProt ID	Gene Name	Protein Name	Abundance Ratio SK-BR3.rTP/ SK-BR3	Fold Change	#Unique Peptides	#PSM	Mascot Score
P02763	ORM1	Alpha-1-acid glycoprotein 1	100	6.64	3	7	77
Q8WW33	GTSF1	Gametocyte-specific factor 1	100	6.64	3	6	97
P54652	HSPA2	Heat shock-related 70 kDa protein 2	100	6.64	1	132	2691
Q9NPA8	ENY2	Transcription and mRNA export factor ENY2	100	6.64	1	5	112
Q99595	TIMM17A	Import inner membrane translocase Tim17-A, mitochondrial	100	6.64	1	4	76
P02790	HPX	Hemopexin	100	6.64	8	9	86
P13637	ATP1A3	Sodium/potassium-transporting ATPase subunit alpha-3	100	6.64	1	104	1896
Q9UNX3	RPL26L1	60S ribosomal protein L26-like 1	100	6.64	1	31	533
Q5T280	SPOUT1	Putative methyltransferase C9orf114	100	6.64	1	3	76
P31151	S100A7	Protein S100-A7	5.03	2.33	5	16	224
P12277	CKB	Creatine kinase B-type	3.95	1.98	11	38	787
Q15050	RRS1	Ribosome biogenesis regulatory protein homolog	3.80	1.93	1	4	78
P04179	SOD2	Superoxide dismutase (Mn), mitochondrial	3.77	1.92	7	39	402
P31327	CPS1	Carbamoyl-phosphate synthase (NH3), mitochondrial	3.76	1.91	28	70	854
P49810	PSEN2	Presenilin-2	3.17	1.67	2	4	93
P17900	GM2A	Ganglioside GM2 activator	3.07	1.62	4	21	174
P25311	AZGP1	Zinc-alpha-2-glycoprotein	2.79	1.48	3	3	92
Q06787	FMR1	Synaptic functional regulator FMR1	2.72	1.45	1	11	173
Q9Y3B8	REXO2	Oligoribonuclease, mitochondrial	2.66	1.41	1	8	95
Q13011	ECH1	Delta(3,5)-delta(2,4)-dienoyl-CoA isomerase, mitochondrial	2.62	1.39	8	79	1337
Q8NEJ9	NGDN	Neuroguidin	2.60	1.38	3	11	189
P23434	GCSH	Glycine cleavage system H protein, mitochondrial	2.57	1.36	2	7	129
Q6UB35	MTHFD1L	Monofunctional C1-tetrahydrofolate synthase, mitochondrial	2.52	1.33	7	22	218

**Table 4 ijms-25-00207-t004:** List of the top proteins identified via label-free MS/MS as being less abundant in the SK-BR-3.rTP cell line (Mascot score > 70, at least one unique peptide, a q-value < 0.05, an abundance ratio variability < 30%, and an abundance ratio < 0.4). An abundance ratio = 0.01 means that none of the peptides identified in the sensitive parental cell samples were found in their corresponding resistant ones.

UniProt ID	Gene Name	Protein Name	Abundance Ratio SK-BR3.rTP/ SK-BR3	Fold Change	#Unique Peptides	#PSM	Mascot Score
Q9UHD9	UBQLN2	Ubiquilin-2	0.01	−6.64	2	13	138
Q01814	ATP2B2	Plasma membrane calcium-transporting ATPase 2	0.01	−6.64	1	12	95
P55212	CASP6	Caspase-6	0.01	−6.64	2	6	145
Q92626	PXDN	Peroxidasin homolog	0.01	−6.64	3	6	109
A8MXV4	NUDT19	Acyl-coenzyme A diphosphatase NUDT19	0.01	−6.64	3	7	142
O76070	SNCG	Gamma-synuclein	0.01	−6.64	2	3	79
P47224	RABIF	Guanine nucleotide exchange factor MSS4	0.01	−6.64	1	6	78
Q15654	TRIP6	Thyroid receptor-interacting protein 6	0.07	−3.92	14	29	272
P43155	CRAT	Carnitine O-acetyltransferase	0.19	−2.38	5	9	98
P15153	RAC2	Ras-related C3 botulinum toxin substrate 2	0.20	−2.31	1	17	226
P80365	HSD11B2	Corticosteroid 11-beta-dehydrogenase isozyme 2	0.21	−2.27	3	7	154
O43570	CA12	Carbonic anhydrase 12	0.24	−2.09	1	6	98
Q9NVA2	SEPTIN11	Septin-11	0.26	−1.95	7	36	492
P05161	ISG15	Ubiquitin-like protein ISG15	0.27	−1.91	2	4	94
Q12800	TFCP2	Alpha-globin transcription factor CP2	0.28	−1.83	3	7	152
Q9BS40	LXN	Latexin	0.29	−1.80	3	10	172
P40121	CAPG	Macrophage-capping protein	0.29	−1.79	9	45	773
Q8NEB7	ACRBP	Acrosin-binding protein	0.30	−1.75	1	21	372
P09758	TACSTD2	Tumor-associated calcium signal transducer 2	0.31	−1.70	5	23	377
O94992	HEXIM1	Protein HEXIM1	0.31	−1.70	2	10	162
P04083	ANXA1	Annexin A1	0.32	−1.65	10	28	426
P18564	ITGB6	Integrin beta 6	0.33	−1.59	3	5	103
P05165	PCCA	Propionyl-CoA carboxylase alpha chain, mitochondrial	0.34	−1.57	7	10	144
Q8IW45	NAXD	ATP-dependent (S)-NAD(P)H-hydrate dehydratase	0.37	−1.44	5	10	163
Q96AY3	FKBP10	Peptidyl-prolyl cis/trans isomerase FKBP10	0.38	−1.41	14	70	998
Q9NQ29	LUC7L	Putative RNA-binding protein Luc7-like 1	0.38	−1.40	1	14	131
Q08AI8	MAB21L4	Protein mab-21-like 4	0.38	−1.38	6	13	172
Q5VWZ2	LYPLAL1	Lysophospholipase-like protein 1	0.39	−1.36	3	9	120
O96011	PEX11B	Peroxisomal membrane protein 11B	0.39	−1.36	5	11	193
13796	LCP1	Plastin-2	0.39	−1.34	18	92	1319

## Data Availability

The MS proteomics data have been deposited to the ProteomeXchange Consortium via the PRIDE partner repository with the dataset identifier PXD045804.

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
