# Peer review of "Generation and Characterization of Trastuzumab/Pertuzumab-Resistant HER2-Positive Breast Cancer Cell Lines"

_ijms, 2023, doi:10.3390/ijms25010207_

Round 1

Reviewer 1 Report

Comments and Suggestions for Authors

In this manuscript, the authors have generated and characterized four trastuzumab/pertuzumab-resistant HER2-positive breast cancer cell lines. Overall, the reported results are interesting, and the conclusion of the manuscript is well supported. I recommend accepting this manuscript with a minor revision.

1.                    The authors can add a wound-healing assay to support their data and to access the migration/metastatic potential of the resistant cells —a comparison between treated and untreated cells.

2.                    Why didn’t the authors analyze the resistant cells' gene expression pattern to discover significantly affected genes to be used as markers?

3.                    It would also be interesting to test other chemotherapeutics drugs, for example, Doxorubicin and Mitoxantrone, to discover if the pathways to drug resistance are the same. I believe the newly developed resistant cells will also be resistant to other drugs as the MAPK pathway is usually over-expressed in cancer cell lines, contributing to drug resistance.

4.                    I am also curious to know if the authors witness any change in the shape and morphology of the trastuzumab/pertuzumab-resistant HER2-positive breast cancer cell lines. The authors have already indicated in their proteomics study that there was dysregulation of the maintenance of the cellular structures. Does this dysregulation contribute in any way to the change in cellular morphology?

Reviewer 2 Report

Comments and Suggestions for Authors

In the manuscript titled ‘ Generation and characterization of trastuzumab/pertuzumab-resistant HER2-positive breast cancer cell lines’ the authors focused on understanding mechanisms of resistance to first-line therapy such as trastuzumab plus pertuzumab in HER-positive breast cancer cells. Since most initial therapies in cancer lead to development of resistance and eventual relapse this study is pertinent and addresses important questions with a time and tested approach of generating resistant cell lines

The authors generated four novel HER-2 positive cell lines by prolonged exposure to T+P and determined their resistance rates and mechanisms.

Overall, I enjoyed the overall flow of the manuscript and approach of understanding resistance mechanism to combinatorial therapy. I have some comments/ questions which I have listed below:

(1)   Fig 1. The authors used prolonged exposure to generate resistant cell lines, 8 months for AU-565, EFM-192A, SK-BR3 and 15 months for BT-474 and acute exposure for generation of acquired resistant lines. The authors used 15ug/ml of trastuzumab + 20ug/ml pertuzumab ( line 293) without any reference to why those concentrations were relevant here? Did they do a prior titration or refer previous work? It would be good to cite that information or address it? They also mention Line 301 they isolated single cell clones. Is the data in Fig3 from the pool population or average of isolated single cell clones?

(2)    It wasn’t clear which stage the results in Fig1 were calculated. Is that 7 day exposure towards the end of the 8 month/15 month exposure?

(3)   They mention about long-term resistance development where the cells (line 327) were cells treated for 21 days with 1ug/ml T and 1.3ug/ml of P. Again no mention of why those concentrations were relevant and how they chose it?

(4)   It’s a little confusing how the acquired resistance and long-term resistance were developed although it mentions long term was prolonged exposure to 15ug/ml T and 20ug/ml P for 8/15 months whereas there is a mention of a lower dose exposure as well?

(5)   BT-474 seems to be the most sensitive cell line out of all. It would be nice to address any differences in its resistance mechanisms vs AU-565 which is the least sensitive to T+P treatment? Is it only the increased expression of phosphorylated forms of the four HER receptors that’s leading to resistance? ( line 410?) Why is that not the same for the other 3 lines? What’s causing the difference? What gene amplifications are already present in these lines?

(6)   In the cell line authentication and evaluation of point mutation qPCR assay in Table2 ( line 366) the authors did not clarify which specific mutations in NFKB2, TP53, SMAD4 and NOTCH2 were evaluated ( just mention from CCLE)? It’s also not clear how many single cell clones were evaluated here for the sensitive parental vs resistant cell lines? I do not see much value generated with the results shown here as its pretty vague? I would like to see the specific mutations assayed and also number of clonal cell lines used in these assays? It would have been nice to incorporate data from COSMIC (Catalogue of Somatic Mutations In Cancer), cBioPortal, TCGA , CCLE, relevant to breast cancer specific mutations that they tested here. Message is not very clear.

(7)   In Fig 4 panel C although pHER3 levels are reported as 0.91 in EFM-192A T+P and 0.75 EFM-192ArTP I did not notice any visible bands, so not sure about the accuracy of the densitometric analysis? What is causing this visual difference? Saw similar trends in Fig5 as well. Panel 5B BT474 , C and T+P for pP38  no bands visible although levels were reported at 1.00 and 1.04

(8)   Not very clear why the proteomics data was collected on only one Her-2 positive line SK-BR-3. It would make sense to generate for AU-565 as well. What is the HER-2 amplification status of the two Her-2 positive lines and was it different between the lines?

Overall, I am partly convinced by some of the results. The clone selection was not very clear to me and also the western blot results and conclusions were nebulous. Maybe comparison of Her-2 positive cancer vs luminal types would help improve the overall message of disease resistance mechanisms?

Reviewer 3 Report

Comments and Suggestions for Authors

This manuscript described the generation of trastuzumab/pertuzumab-resistant HER2-positive breast cancer cells. Overall, it is well structured and easy to follow. I read the manuscript with great interest. More optimization should be given before it can be accepted for publication. The detailed comments can be found below:

1. On Page 4 Line 154, I wonder whether the dose of 15 g/mL or 20 g/mL of trastuzumab or pertuzumab was too high for these breast cancer cells after resistance establishment.

2. On Page 11 Line 355, some explanations should be given on the little differences in HER2 expression after the acquisition of resistance.
